# Fluorescent Molecularly Imprinted Polymers Loaded with Avenanthramides for Inhibition of Advanced Glycation End Products

**DOI:** 10.3390/polym15030538

**Published:** 2023-01-20

**Authors:** Pei Zhu, Ying Zhang, Dianwei Zhang, Huilin Liu, Baoguo Sun

**Affiliations:** 1School of Food and Health, Beijing Technology and Business University, Beijing 100048, China; 2School of Light Industry, Beijing Technology and Business University, Beijing 100048, China

**Keywords:** molecular imprinted polymers, avenanthramides, reverse microemulsion, release, pyrraline

## Abstract

Encapsulating bioactive avenanthramides (AVAs) in carriers to respond to the environmental changes of food thermal processing allows the controlled release of AVAs for the effective inhibition of biohazards. In this study, fluorescent molecular imprinted polymers (FMIPs) loaded with AVAs were prepared by reverse microemulsion. The fluorescent signal was generated by carbon dots (CDs), which were derived from oat bran to determine the load of AVAs. The FMIPs were uniformly spherical in appearance and demonstrated favorable properties, such as thermal stability, protection of AVAs against photodegradation, high encapsulation efficiency, and effective scavenging of free radicals. After consideration of the different kinetics models, the release of AVAs from the FMIPs matched the Weibull model and followed a Fickian diffusion mechanism. The FMIPs exhibited good inhibition of pyrraline in a simulated casein-ribose system and in milk samples, indicating the release of AVAs could inhibit the generation of pyrraline.

## 1. Introduction

Advanced glycation end products (AGEs) are generated with the non-enzymatic browning reaction of reducing sugars and protein amino groups during industrial processes, home cooking, and the long-term storage of foods [1]. It is generally accepted that the main compounds of AGEs are pathophysiologically derived from the ingestion of dietary *N*^ε^-carboxymethyllysine (CML), *N*^ε^-carboxyethyllysine (CEL), and pyrraline (PRL). The glycosylation of dietary proteins in food processing may lead to the formation of potentially dangerous AGEs, which will cause certain harm to human health. Therefore, there is widespread research into the development of inhibitors that can inhibit the formation of AGEs.

Polyphenols are natural inhibitors that have been reported to show good inhibition effects against AGEs [2]. Quercetin can effectively inhibit the formation of AGEs in a dose-dependent manner via trapping reactive dicarbonyl compounds [3]. The phenolic compounds extracted from highland barley by Zhang et al. are able to inhibit PRL production [4]. Avenanthramides (AVAs) are a unique phenolic alkaloid in oats that were originally identified as phytoantitoxins produced by plants with potential health promoting properties [5]. In structural terms, AVAs contain a part of anthranilic acid that is partially bound to phenylenic acid by an amide bond [6]. AVAs have good antioxidant activity in vitro and good therapeutic effects on atherosclerosis, inflammation, diabetes, and tumor diseases [7]. Interestingly, AVAs only exist in oats, with more than 20 different forms. The three main forms are AVAs 2p, AVAs 2c, and AVAs 2f [8]. Our groups study the inhibitory effect of AVAs on AGEs and find that AVAs have the ability to scavenge free radicals to inhibit AGEs, but these polyphenol compounds have the common characteristics of instability against light and thermal [9]. Therefore, the encapsulated AVAs in the carriers during the thermal processing of food can protect it from decomposition. When certain conditions are reached, AVAs will be released so as to achieve the efficient inhibition of AGEs.

Molecular imprinting technology (MIT) is a kind of molecular recognition research that uses natural compounds or synthetic compounds to simulate a biological system. This system is used to synthesize molecularly imprinted polymers (MIPs) that have specific recognition and selective adsorption properties [10]. MIPs are most widely used in biosensors [11,12,13,14], catalysis [15], antibody simulation [16], molecular recognition [17,18], drug delivery [19,20], and bio-isolation, as well as in diagnostic and therapeutic applications [21,22]. In addition to these functions, MIPs can also absorb a large number of substances that are structurally similar to the imprinted molecule, which is used as a reaction-controlled release carrier [23]; He et al. prepared pH-sensitive MIPs containing adenosine triphosphate by thermally induced polymerization and could release the target molecules well [24]. Similarly, Ding et al. discovered that mesoporous silica could be selected as the ideal molecular imprinting matrix candidates [25]. Song et al. prepared molecularly imprinted polymer (MIP) microspheres with porous structures that were prepared by a combined suspension-iniferter polymerization method using capecitabine (CAP) as a template molecule [26]. In addition, magnetic molecularly imprinted polymers (MMIPs) have high selectivity in sample pre-treatment and allow for the fast and easy isolation of the target analyte [27].Unfortunately, MIPs have no signal response capability and cannot accurately show the content of enriched targets. Fluorescent MIPs (FMIPs) have the high specificity of molecular imprinting and the signal response ability of fluorescence intensity, so they can visually display the content of the target. It is usually composed of a core-shell structure with a fluorescent luminous center as the core and MIPs wrapped outside. Carbon dots (CDs) are used as common fluorescence centers with the advantage of good solubility, stability, biocompatibility, and low toxicity [28]. The popularization of green synthesis technology endows the CDs with green precursors, prompting people to have a strong interest in green CDs synthesis [29].

In the study, we prepared FMIPs loaded with AVAs 2f that could be used for inhibiting PRL in milk processing. Here, CDs derived from oat bran were used as the fluorescence response element to calculate the load of AVAs 2f, which was encapsulated into the FMIPs by reverse microemulsion synthesis. The morphology, structure, and thermal stability of the FMIPs were studied by scanning electron microscopy (SEM), transmission electron microscope (TEM), Fourier transform infrared spectroscopy (FT-IR), X-ray photoelectron spectroscopy (XPS), and thermogravimetry (TG). Furthermore, the impacts of the temperature environment on AVAs 2f release were also investigated by zero order, first order, Higuchi, Weibull, and Hixson-Crowell kinetic models. In this study, the kinetics and mechanisms of AVAs release from FMIPs were, respectively, examined under the simulated conditions of gastrointestinal digestion and food processing. The inhibitory effect on PRL were studied by the simulated casein-ribose system and then applied to authentic milk samples.

## 2. Materials and Methods

### 2.1. Chemicals and Apparatus

Triton X-100 (9002-93-1), cyclohexane (110-82-7), 3-aminopropyltriethoxysilane (APTES, 919-30-2), ethyl silicate (tetraethyl orthosilicate; TEOS, 78-10-4), acetone (67-64-1), and AVAs (108605-69-2) were purchased from Darwin Reagent Co., Ltd., (Beijing, China). The ultrapure water was purchased from the A.S. Watson Group (Hong Kong, China). The SEM images were acquired using an SU 8020 microscope (Hitachi, Tokyo, Japan). The TEM images were recorded using a FEI Tecnai G2 F30 microscope (FEI; Hillsboro, OR, USA) at an accelerating voltage of 200 kV. The FT-IR was collected by a Vertex 70 spectrometer (Bruker, Bremen, Germany). The ultraviolet spectrophotometry was performed using a Cary 100 UV-Vis spectrometer (Agilent, Santa Clara, CA, USA). The fluorescence characteristics were recorded with a Biotek Synergy H1 Multi-Mode Microplate Reader (BioTek; Winooski, VT, USA). 

### 2.2. Design and Synthesis of CDs

The CDs were prepared by a one-step hydrothermal method [30]. The processed naked oats (6 g) were dissolved in distilled water (50 mL) with stirring. The mixed solution was added to 100 mL of a Teflon-lined high-temperature reaction kettle, heating at 200 °C for 3 h. After the mixture cooled to room temperature, it was centrifuged for 10 min at 10,000× *g* rpm. To further remove the remaining particles, the supernatant was filtered through a 0.22 μm microporous membrane. Finally, the CDs solution was conducted through a dialysis membrane (MWCO, molecular weight cut-off: 1000 Da) for 48 h (changed deionized water every 3 h). After this process, a light brown CDs aqueous dispersion was obtained. Then, the aqueous dispersions of the CDs were evaporated to dryness and the crystalline solid was ground to obtain the fluorescent CDs powder.

### 2.3. Preparation of FMIPs 

The FMIPs with different loading contents of AVAs 2f were prepared according to the following methods. The cyclohexane (7.5 mL) was added into a 25 mL round-bottom flask before the Triton X-100 (1.8 mL) was slowly added dropwise with agitation. After stirring for 15 min, the TEOS solution (50 μL), ammonia solution (100 μL), and fluorescent CDs (1 mg/mL, 1 mL) were added to the mixture, respectively. The top of the round-bottom flask was sealed with a sealing film and stirred vigorously for 2 h under strict light protection. Then, APTES (25 μL) and AVAs 2f (template molecules, 4 mg/mL, 100 μL) were added dropwise. Finally, the flask was sealed with tin foil and stirred for 24 h. When the reaction was complete, acetone (10 mL) was added to the reaction system to terminate the microemulsion state, and then the excess impurities and solvents were removed by centrifugation (1000× *g* rpm, 5 min). Furthermore, 10 mL of acetone was added to terminate the reaction and excess solvents, and impurities were removed by centrifugation. Finally, the white precipitate was dried in a vacuum oven at 60 °C before being fully ground to a powder. As a reference, a control (FMIPs without AVAs 2f) was prepared following the same procedure, but without the addition of AVAs 2f. In order to maximize the selectivity and sensitivity of FMIPs to AVAs 2f, the mole ratios of template molecules: APTES: TEOS were varied (1:600:1000, 1:600:1200, 1:600:1600, 1:800:1000, 1:800:1200, 1:800:1600) to establish the optimal ratio according to the observed imprint factor (IF). The linear relationship between the luminescence response to FMIPs and the concentration of AVAs 2f were calculated using the Stern-Volmer equation [31]:(1)$$\frac{F_{0}}{F}=K_{\mathrm{sv}}Q$$
where *F*_0_ and *F* are the fluorescence intensities of FMIPs before and after adding the target, *K*_SV_ is the Stern-Volmer constant (usually refers to the slope), and Q is the concentration of AVAs 2f. The IF was calculated by the ration of *K*_sv,MIPs_ and *K*_sv,NIPs_.

### 2.4. Efficiency of AVAs-2f Encapsulation into FMIPs

In order to obtain the encapsulation efficiency (EE), we added different amounts of AVAs 2f (1, 2, 3, 4, and 5 mg/mL) into the FMIPs, which were called FMIPs@1, FMIPs@2, FMIPs@3, FMIPs@4, and FMIPs@5, respectively. Then, the FMIPs (10.0 mg) were mixed with 1 mL of methanol with shaking, and ultrasonicated in an ice bath for 30 min, and then centrifuged (11,000× *g* rpm) for 10 min at 4 °C. The supernatant was filtered through a 0.45 μm microporous membrane and determined by high-performance liquid chromatography (HPLC) [9,32]. 

The HPLC chromatographic conditions were set as follows: a reversed-phase C18 column (4.6 × 250 mm, 5 µm, Thermo Fisher Technologies Co., Ltd., Waltham, MA, USA) using a linear gradient. Two solvents were used for the mobile phase: solvent A was 0.1% formic acid in water; solvent B was 0.1% formic acid in methanol. Gradient elution was initiated with isocratic elution at 35% B for 2 min, followed by a linear increase to 55% B (2–30 min), and a linear increase to 100% B (30–32 min), and then isocratic elution at 100% B (32–40 min). The sample injection volume was 5 μL and the eluent flowrate was 1.0 mL/min. AVAs 2f was quantified at the UV wavelength of 319 nm, which was the maximum absorption wavelength of AVAs 2f. Before detecting the content of AVAs 2f in the FMIPs mixture, we should use the same method to detect the standard of AVAs 2f by different concentrations. Then, the calibration curves of the concentration and peak areas of AVAs 2f were plotted for the following experiment (Appendix A). The EE of the AVAs 2f was calculated using the following equation:(2)$$\mathrm{Encapsulation}\;\mathrm{efficiency}\;(\%)=\frac{W_{s}}{W_{t}}\times 100\%$$
where W_s_ is the weight of the injected AVAs 2f in the supernatant and W_t_ is the mass of the total AVAs 2f input.

### 2.5. Determination of DPPH Radical Scavenging Activity

A standard solution of DPPH (2,2-diphenyl-1-picrylhydrazyl) with a mass concentration of 0.050 mg/mL was obtained by accurately dissolving 12.5 mg of DPPH standard in 250 mL of anhydrous ethanol [33]. The solution was properly diluted to make the absorbance 0.70 ± 0.05 as the DPPH solution before use. Then, the DPPH solution (1 mL) and the different concentrations of FMIPs-methanol mixture (1 mL) were mixed for 30 min, respectively. Aminoguanidine (AG) was used as the positive control. The absorbance of each solution was determined at 517 nm. In total, 1 mL of anhydrous ethanol was used to replace the sample for the initial determination under the same conditions. The experiment was measured three times in parallel, then the ability of free radical scavenging activity was calculated according to the equation:(3)$$\mathrm{DPPH}\;\mathrm{scavenging}\;\mathrm{activity}\;(\%)=\frac{A_{0}-A}{A_{0}}\times 100\%$$
where A_0_ is the initial concentration of DPPH and A is the absorbance of DPPH after adding the sample solution.

### 2.6. Fluorescence Stability Performance

The stability of FMIPs under long-term ultraviolet irradiation was studied due to AVAs 2f easily decomposing under UV light. The FMIPs were dispersed in a mixture of methanol/water (1:1, *v/v*) to prepare the mixture (1 mg/mL). Then, the mixture was stored for 12 h at room temperature under a 36W UV lamp (254 nm), and the concentrations of AVAs 2f in different groups were detected by HPLC.

### 2.7. In Vitro Release of AVAs 2f

The in vitro release of FMIPs was tested by dynamic dialysis [34]. The FMIPs (5 mg) were dispersed in 2 mL of the methanol solution and then transferred to a pre-activated semi-permeable membrane dialysis bag (molecular weight cut-off, MWCO, 8 kDa). The dialysis bag was clamped and fixed at both ends. The test dialysis bag was placed in a brown bottle with 50 mL of phosphate buffered saline (PBS, PH = 7.4), and stirred with a magnetic stirring apparatus. Then, the system temperature was set at 37 ± 0.5 °C. Similarly, by simulating the release of FMIPs during food processing, we selected 60 °C and 80 °C for simulated release. To assess the release of FMIPs at these temperatures, the experiments were repeated at these temperatures following the same process. The experiments were carried out under continuous agitation (200 rpm). At different time points (0.25, 0.5, 1, 2, 4, 8, 12, 24, 36, 48 h), an aliquot of solution (2 mL) was extracted each time, and the same volume of PBS was supplemented. The extracted samples were detected by HPLC and the cumulative release (Q) of AVAs 2f was calculated according to the following formula:(4)$$Q\;(\%)=\frac{C_{n}v+V_{i}\sum_{i=1}^{n-1}C_{i}}{m}\times 100\%$$
where C_i_ and C_n_ are the concentrations at different time points, v and V_i_ are the PBS solution (50 mL) and sample volume (2 mL), respectively, and m is the initial amount of the AVAs 2f.

### 2.8. Preparation of Casein-Ribose Simulation System

To further test the inhibitory effect of FMIPs on PRL, we prepared a simulated system of casein-ribose to verify the feasibility of FMIPs. The *β*-casein (3 g) and D-ribose (2.7 g) were dissolved in PBS (100 mL, pH = 7.4) and the mixture was stirred at room temperature for 10 h until the *β*-casein was completely dissolved. We transferred 6 mL of the solution into a centrifuge tube and then added FMIPs (1 mg/mL, 1 mL) into the treated simulation system solution (1 mL). The mixture was heated at 60 °C or 80 °C for 30 min, respectively. The control group was simulated the system solution without the FMIPs.

### 2.9. Detection of PRL by HPLC

The HPLC chromatographic conditions for PRL detection were as follows. Separation was performed on an Inertsil ODS-SP column (250 mm × 4.6 mm, 5 μm). The mobile phase A was 0.1% trifluoroacetic acid (TFA) and 99.9% water, and mobile phase B was acetonitrile. The mixing ratio of mobile phases A and B were mixed at 1:1 (*v/v*), and the flow rate was 1 mL/min. The analysis was executed using an injection volume of 10 μL and the detection wavelength was 297 nm [35]. All the solvents were filtered using a 0.22 μm membrane filter (Nylon film, suitable for organic solution filtration) before being detected by HPLC.

### 2.10. Testing of Milk Samples

The milk purchased from a market was used as the authentic sample and the FMIPs as the inhibitors to conduct the experiments. Firstly, the milk (2 mL) was heated at 60 °C or 80 °C for 30 min and added to a 50 mL centrifuge tube, and then 20 mL of methanol (containing 5% ammonia) was added. The homogenizer was used to stir the milk into the homogenate, and the ultrasonic treatment was conducted (30 min, 80 KHZ). After the solution was taken to dryness by the rotatory evaporator, it was re-dissolved in 2 mL of ultrapure water waiting for HPLC analysis. Then, the control group (added 1mg/mL FMIPs) and the positive control group (added 1 mg/mL aminoguanidine) were prepared in the same way. Finally, after the reaction, we analyzed each sample by HPLC to determine the inhibitory effect of FMIPs on PRL in the actual sample.

### 2.11. Statistical Analysis 

All data were conducted in triplicate, and the results were expressed as the mean values ± standard deviation. The analysis of the variance was performed to determine the statistically significant differences between the groups. The statistical differences of the data were analyzed by one-way analysis using IBM SPSS Statistics 25.0. The figures were drawn by Origin 2018 software.

## 3. Results

### 3.1. Morphological and Structural Characterization of FMIPs

The process of FMIPs preparation and the load-releasing mechanism of AVAs 2f were shown in Figure 1. In the presence of the crosslinking agent, APTES and AVAs 2f recognized each other through hydrogen bonding and crosslinked to form polymers on the surface of the CDs. After the template molecule was released by washing the template, three-dimensional pores and specific recognition sites with specific spatial orientation were formed on the surface of FMIPs, which could specifically recognize AVAs 2f. Moreover, the released AVAs 2f had an inhibitory effect on the AGEs in the milk at different temperatures. TEOS and APTES hydrolyzed and condensed on the CDs’ surface, which coated the CDs’ surface with a silica layer and formed amino groups on the surface for the subsequent reaction. The added amount of the functional monomer had a significant impact on the imprinting efficiency and affinity for AVAs 2f in FMIPs, while the molar ratio of the template molecule to APTES affected the number of recognized sites in FMIPs. Therefore, six molar ratios (1:600:1000, 1:600:1200, 1:600:1600, 1:800:1000, 1:800:1200, and 1:800:1600) were used to determine the optimal molar ratio of the template molecule, functional monomer, and crosslinker by comparing the IF. The results (Figure 1A) showed that the maximum IF (3.34) was obtained for AVAs 2f with a molar ratio of 1:800:1000. This ratio appeared to provide an adequate amount of TEOS for the FMIPs to form a rigid porous structure with specific recognition sites. At the same time, the thick shell structure generated by the excessive TEOS (molar ratio greater than 1:800:1000) should be avoided because a thick-walled structure would reduce the fluorescence intensity of the material and hinder the adsorption of AVAs 2f. When the molar ratio of AVAs 2f: APTES: TEOS was 1:800:1000, the FMIPs had the maximum adsorption selectivity for AVAs 2f as the target substance. Then, the loading content of AVAs 2f for different imprinted polymer samples was calculated to be 4.9–38.1%. Among them, FMIPs@4 had the highest encapsulation efficiency (Appendix A). The apparent morphologies of FMIPs and FMIPs without AVAs 2f were observed by high-resolution SEM and TEM. Figure 1B accurately represented the content of the major elements in FMIPs, among which the C element had the highest content, and the N element had the lowest content, and the presence of the Si element provided further confirmation of the formation of a SiO_2_ layer. SEM analysis of FMIPs (Figure 1C) showed a closely arranged structure of uniformly sized round spheres. The morphologies of the CDs were observed by TEM, and the CDs were close to spherical (Appendix A). The average size of the CDs was 12 nm (Appendix A), which was calculated using particle size distribution analysis software (Nano Measurer 1.2). As observed in Figure 1D, the CDs were wrapped by FMIP layers and the core-shell structure was apparent. In addition, FMIPs had the appearance of porous microspheres, being evenly dispersed with rough surfaces. The illustration in the upper right corner of Figure 1D showed that the average particle size of FMIPs was between 70 and 80 nm. Furthermore, the morphology of FMIPs without AVAs 2f was similar to that of the core-shell nanospheres, although the spheres of FMIPs without AVAs 2f had more imprinting sites for AVAs 2f (Appendix A). To further determine the encapsulation of AVAs 2f, we performed element mapping analysis using energy dispersive spectroscopy (EDS). As shown in Appendix A, the EDS of FMIPs without AVAs 2f show that the distribution of C, N, O, Si elements were similar to the FMIPs, which illustrated that the construction of the FMIPs without AVAs 2f was similar to the FMIPs. As shown in Figure 1E and Appendix A, the FMIPs contained more C, O, N, and other elements and were more concentrated in the ball than those FMIPs without AVAs 2f, which provided evidence for the successful encapsulation of AVAs 2f. 

The chemical bonds and functional groups of FMIPs, FMIPs without AVAs 2f, CDs, and AVAs 2f were characterized by FT-IR spectroscopy. As shown in Figure 2A, CDs had a characteristic hydroxyl band at 3376 cm^−^^1^, while the peak appears at 2877 cm^−^^1^ and was assigned to the C-H vibration of methylene. However, the characteristic peaks of the CDs at 1651 cm^−^^1^ and 1361 cm^−^^1^ disappeared after polymer synthesis, indicating that the CDs reacted with functional monomers of APTES. The specific peaks of the FMIPs were C-H stretching vibrations of the aromatic ring from the benzoate fraction at 2901 cm^−^^1^ and Si-O-Si bond at 1020 cm^−^^1^, respectively. The results showed that alkyl decomposition, intermolecular cyclization, and condensation reactions occurred during the formation of the polymer. The specific peaks of AVAs 2f at 3012 cm^−^^1^ and 2987 cm^−^^1^ were due to the tensile vibration of aromatic C-C-H and C-H. The peaks at 1689 cm^−^^1^ and 1032 cm^−^^1^ were attributed to the skeleton vibration of the benzene ring and the stretching vibration of C-O and benzene ring skeleton. However, the characteristic peak content of the FMIPs without AVAs 2f at 2901 and 1020 cm^−^^1^ was lower than that of the FMIPs. These characteristic peaks were observed in both AVAs 2f and the FMIPs, thus demonstrating that the shell wall material successfully encapsulated AVAs 2f. The advantage of XPS was that it could not only detect the chemical composition of the surface, but also determine the chemical state of each element. Figure 2B showed the XPS spectrum of FMIPs and at 150 and 180 eV, and it represented Si_2p_ and Si_2s_. The high-resolution C_1s_ spectrum (Figure 2C) showed three types of carbon bonds: C-C/C=C (C_1_), C-O (C_2_), and C=O (C_3_), with high-resolution peaks of 284.5, 286.2, and 288.1 eV, respectively. These three component peaks were classified according to the types of bonds in which C participates in the structure of the shell wall itself. The deconvolution of O_1s_ spectrum (Figure 2D) corresponded to C=O and C-O with peak values of 531.9 eV and 532.5 eV. The N_1s_ spectrum shown in Figure 2E showed the formation of amides associated with the functional groups -N-H- and C-N. The XPS data of the CDs was shown in Appendix A, which was similar to the FMIPs. The above analysis showed that the FMIPs could be prepared by reverse microemulsion synthesis and successfully loaded with AVAs 2f.

### 3.2. Thermogravimetric Analysis of FMIPs

Thermogravimetric analysis (TGA) was performed to determine the relationship between thermal stability and temperature for the FMIP. TGA curves of FMIPs, FMIPs without AVAs 2f, and CDs were shown in Figure 3A. From these TGA curves, the initial weight loss of the CDs occurred at around 180 °C, while this trend occurred for the FMIPs and FMIPs without AVAs 2f at around 400 °C. These results were attributed to the CDs in the recognition active sites in the cavities, which were highly sensitive at higher temperatures. In contrast, the presence of hydrogen bonds in the FMIPs led to higher thermal stability. When the temperature rose to 600 °C, the quality of the CDs decreased by 47.4%. When the temperature increased further to 950 °C, its mass decreased by 61.5%. There was basically no difference between the TGA curves of FMIPs and FMIPs without AVAs 2f, indicating that the thermal stability was not related to load AVAs 2f, but rather to the composition of the FMIPs. The result was further proved by differential thermogravimetry (DTG) analysis (Figure 3B); in the high and low temperature region, the DTG curve of FMIPs was similar to the FMIPs without AVAs 2f, which further proved that the FMIPs had good thermal stability. 

### 3.3. DPPH Radical Scavenging Analysis 

DPPH was commonly used for in vitro screening of antioxidants because it had a stable lone electron structure. When mixed with the antioxidant, DPPH free radical accepting electrons fade, while the antioxidant loses electrons, and the corresponding UV absorption decreased or even disappeared. According to this principle, we used UV spectrophotometry to measure the change of the absorbance value to judge the scavenging ability of the FMIPs on DPPH free radical. As shown in Figure 3C, AG was used as a positive control to evaluate the antioxidant capacity of the system. Within the concentration range of 0.2–1.0 mg/mL, the scavenging capacity of FMIPs on DPPH free radical increased with the increasing concentration, which illustrated that AVAs 2f was released from FMIPs to combine with DPPH free radical. Therefore, FMIPs had the ability of scavenge DPPH radical, and the reverse microemulsion technology could be used as a feasible method to improve the natural antioxidant capacity of AVAs 2f.

### 3.4. Fluorescence Stability

AVAs 2f was a phenolic alkaloid that could effectively inhibit AGEs, but it was easy to decompose under UV light and high temperature. Considering the prepared core-shell structure of the prepared FMIPs, it could prevent the interference of light and high temperatures on AVAs 2f. As shown in Figure 3D, free AVAs 2f degraded rapidly under the UV light (254 nm, 36 W). During the first 4 h, 80.73% of the free AVAs 2f was lost. In contrast, the degradation rate of the FMIPs loaded with AVAs 2f was only 22.03% after UV irradiation for 12 h, because AVAs 2f was protected by the FMIPs nanocarrier. Through these results, we confirmed that the FMIPs were an excellent nanocarrier to improve the UV resistance of AVAs 2f. Therefore, this system had great potential for application in food processing.

### 3.5. AVAs-2f In Vitro Release

The in vitro release of FMIPs loaded with AVAs 2f was studied by dynamic dialysis, and the release of AVAs 2f in the gastrointestinal tract and food processing was simulated with a self-made device. Three different temperatures (37, 60, and 80 °C) were set and, respectively, measured for 48 h under the same experimental conditions. The results of the cumulative release were shown in Figure 4A. The increase of temperature significantly improved the release rate of AVAs 2f from the FMIPs. After the FMIPs and release medium were incubated at 37, 60, and 80 °C for 48 h, the cumulative release amount of AVAs 2f reached 58.4%, 77.6%, and 80.2%, respectively. In order to further understand the mechanism of release AVAs 2f from FMIPs, the Weibull, zero order, first order, Higuchi, and Hixson-Crowell release models were examined (Figure 4B–F and Appendix A). The dynamic release model fitting could be analyzed according to the coefficient of association R^2^ and the residual sum of squares (RSS). In the above dynamic model, the correlation coefficient R^2^ represented the correlation, with larger values implying better correlation. The R^2^ values of the Weibull equation (Appendix A) at 37, 60, and 80 °C were 0.9935, 0.9968, and 0.9885, respectively. The R^2^ correlation at 60 °C was the strongest, indicating that the release model might conform to the Weibull model. However, the R^2^ of other models were relatively low, which showed that the sustained release model of AVAs 2f did not fit well with other kinetic models and could not release AVAs 2f well according to its model. In addition, the smaller RSS value implied the higher fitting degree. Among all the simulated equations, the RSS of the Weibull equation was the smallest, which was consistent with the trend for the correlation coefficient R^2^. Therefore, the comparison and calculation proved that the Weibull model was the best fitting theoretical model, whose calculation formula was as follows [36]:(5)$$\mathrm{lnln}\left[100/\left(100-Q\right)\right]=K_{0}\mathrm{lnt}-Q_{0}$$
where Q is the cumulative drug release, Q_0_ is the initial amount of drug in the medium, and K_0_ is the first-order release rate constant. The K_0_ values in the range of 0.75–1.0 indicated that the release followed the diffusion mechanisms (Fickian diffusion and Case II transport). As shown in Figure 4B, the K_0_ value in the dynamics curve of the Weibull model was 0.754 at 37 °C, while the K_0_ value was 0.42 and 0.39 at 60 °C and 80 °C, respectively. Therefore, the release process of AVAs 2f from the FMIPs followed the Fickian diffusion mechanism. The above results indicated that the diffusion effect was the main factor for this release process. It was likely that the increased release of AVAs 2f at the high temperature was caused by the core dissolution of the carrier. When the temperature reached a certain value, the AVAs 2f would be in a free flow state, and the copolymer shell rapidly collapsed, resulting in a fast release rate. It was also possible that the release mechanism was caused by the degradation, cracking, or conformational change of the FMIPs, leading to the enhanced release of encapsulated active ingredient AVAs 2f [37]. These results indicated that the effect of temperature on the release of AVAs 2f from the FMIPs was relatively stable, which would be suitable in the development of applications for FMIPs.

### 3.6. Inhibition of PRL by FMIPs in Simulation System

We evaluated the inhibitory effect of the FMIPs on PRL in the casein-ribose simulation system. Firstly, we prepared a standard curve for PRL by HPLC, which fitted the linear formula: y = 7.8583x − 1.537 (Appendix A). As shown in Figure 5, it was the HPLC chromatogram of PRL in the simulated system without added FMIPs. The peak value generated at 28 min was consistent with PRL standards, and its concentration of PRL was 115.3 μg/L. Figure 5B was the AG control group at 60 °C, and the PRL concentration was 56.96 μg/L. It indicated that the FMIPs significantly reduced the concentration of PRL in the simulated system. By comparison, it could be concluded that FMIPs had a 50.6% inhibition rate on the PRL at 60 °C. The content of PRL was also decreased to 69.61 μg/L at 80 °C, indicating that the FMIPs had a 43.1% inhibition rate on PRL. These results indicated that the FMIPs had a certain inhibitory effect on PRL, and a good releasing effect on AVAs 2f. Therefore, these results provide further confirmation that FMIPs successfully released AVAs 2f to inhibit PRL.

### 3.7. Inhibition of PRL by FMIPs in Milk Samples

The AGEs were produced during the pasteurization processing of the milk samples, and two kinds of fresh milk were selected as authentic samples. After the samples were pretreated and analyzed PRL by HPLC, both milk samples contained a number of PRL. The inhibition of PRL by FMIPs was studied at 60 d for a number of PRL. The released AVAs 2f to inhibit PRL are summarized in Table 1. The content of PRL was 4.62 μg/L and 4.03 μg/L in milk samples, at 60 °C without the addition of the FMIPs. With the addition of the FMIPs into the milk samples, the relatition rate was 36.3% and 44.9% at 60 °C. Similar results were observed at 80 °C, the inhibition effects were 36.3% and 44.9%, respectively. The inhibition of PRL was more obvious in the milk samples by adding the FMIPs compared with AG positive group. According to these results, it could be concluded that the FMIPs had a controlled release performance on AVAs 2f.

## 4. Conclusions

In this study, the FMIPs loaded with AVAs 2f were successfully synthesized by the reverse microemulsion method. The morphological characteristics and formation mechanism of FMIPs were determined, and the loading rate was 38.1% (*w/w*). AVAs 2f released by FMIPs was temperature-sensitive, and the release rate of the low temperature (37 °C) was slower than that of the high temperature (60 °C and 80 °C). The effect of the temperature on the release rate was explained in terms of the diffusion of AVAs 2f being affected by core dissolution of the carrier and a distance limitation mechanism. The release of AVAs 2f followed the Fickian diffusion mechanism, which explained the main factors of AVAs 2f release from the FMIPs. In addition, by evaluating the antioxidant and photostability of FMIPs, it is clear that AVAs 2f could be encapsulated and released to effectively scavenge free radicals well under certain conditions. Through the detection of PRL in the actual samples, it was confirmed that the FMIPs could release AVAs 2f well to inhibit the generation of PRL. Therefore, this study had successfully prepared the FMIPs that could be used in the future food processing. This work will stimulate the further development of new anti-glycosylation inhibitors and a new approach for the treatment of glycosylation-related diseases.

## Data Availability

Data is contained within the article or Appendix A.

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
