# Peer review of "Fluorescent Molecularly Imprinted Polymers Loaded with Avenanthramides for Inhibition of Advanced Glycation End Products"

_polymers, 2023, doi:10.3390/polym15030538_

Round 1
Reviewer 1 Report
Manuscript ‘Fluorescent molecularly imprinted polymers loaded with avenanthramides for inhibition of advanced glycation end products’ describes the development of fluorescent molecular imprinted polymers loaded with bioactive avenanthramides were prepared by reverse microemulsion. Bioactive avenanthramides were successfully synthesized by reverse microemulsion method. The fluorescent signal was generated by carbon dots (CDs), which was derived from oat bran to determine the load of bioactive avenanthramides. The fluorescent molecular imprinted polymers were uniformly spherical in appearance and demonstrated favourable properties, such as thermal stability, protection of bioactive avenanthramides against photodegradation, high encapsulation efficiency, and effective scavenging of free radicals.
Overall this manuscript is well written and is rather interesting from the analytical point of view. Manuscript eventually can be published after some minor improvements:
Recent papers on Molecularly imprinted polymers are not well overviewed. Therefore, some recent reviews and other references on the development of Molecularly imprinted polymers (e.g., electrochemically deposited molecularly imprinted polymer-based sensors) could be considered and overviewed discussed in Introduction part of the manuscript.
Section, which is describing MIP formation and removal of template, could be extended and advanced by adding some more details.
Temperature-controlled release of material is very interesting and promising method, therefore, some discussion on possible multiple reusability of formed MIPs could be added.
Author Response
Response to Reviewer 1
- This work synthesized silica-based MIPs. However, the authors did Insufficient literature research. Some representative silica-based MIP works are not highlighted in this manuscript, like doi.org/10.3390/polym10030298; DOI 10.1088/1361-6528/aace10, etc
Answer: Thanks for your comment. Reference [33] has been added in the revised manuscript on Line 60-62 and shown as below. Thanks again for your comment.
- Ding, Z. Li, Y. Cheng, C. Du, J. Gao, Y. Zhang, N. Zhang, Z. Li, N. Chang, X. Hu, Enhancing adsorption capacity while maintaining specific recognition performance of mesoporous silica: A novel imprinting strategy with amphiphilic ionic liquid as surfactant, Nanotechnology. 29 (37) (2018) 13. https://doi.org/10.1088/1361-6528/aace10.
- It is recommended to provide the TEM characterization of CDs.
Answer: Thanks for your comment. The TEM characterization of CDs has been added to Figure. S3A of the support information and shown as below. Thanks again for your comment.
Figure S3A The TEM characterization of CDs
- What is the imprint factor?
Answer: Thanks for your comment. Imprint factor (IF) is the ratio of the molecularly imprinted polymer to the number of non-imprinted polymer binding templates, and the ability to identify specificity is assessed based on the IF. In the study, the IF was 3.34. Thanks again for your comment.
- English still needs to be polished.
Answer: Thanks for your comment. We have carefully revised and improved the language throughout the manuscript. The corrected details are listed as highlighted in the revised manuscript.
- The system did add any thermal response monomer. Why can it achieve temperature-dependent release? I would rather avoid in the abstract usage of 2c, 2p etc.
Answer: Thanks for your comment. I am very sorry about it, I did not add the thermal response monomer. Because the PRL was produced at body temperature (in vivo), and thermal processing of milk, such as pasteurization, we study the release model at 37, 60, and 80 ℃. Considering the imprecision of temperature-dependent release, we have deleted it in the revised manuscript. I have revised the abstract.

Reviewer 2 Report
Please see the attachment.

Author Response
I think the comment is not our study. Please check it.

Reviewer 3 Report
In this manuscript, the author synthesized a fluorescent molecular imprinted polymer for inhibiting pyrraline in the casein-ribose system and in milk samples. And carbon dots are used to dope in the MIPs for the fluorescent response. The whole paper looks good and well-organized. Hence, I recommend this paper can be accepted after minor revision.
1. This work synthesized silica-based MIPs. However, the authors did Insufficient literature research. Some representative silica-based MIP works are not highlighted in this manuscript, like doi.org/10.3390/polym10030298; DOI 10.1088/1361-6528/aace10, etc
2. It is recommended to provide the TEM characterization of CDs.
3. What is the imprint factor?
4. English still needs to be polished.
5. The system did add any thermal response monomer. Why can it achieve temperature-dependent release?
Author Response
Response to Reviewer 2 comments
- Recent papers on Molecularly imprinted polymers are not well overviewed. Therefore, some recent reviews and other references on the development of Molecularly imprinted polymers (e.g., electrochemically deposited molecularly imprinted polymer-based sensors) could be considered and overviewed discussed in Introduction part of the manuscript.
Answer: Thanks for your comment. References [34] and [35] have been added in the revised manuscript on Line 64-66 and shown as below. Thanks again for your comment.
- Song, J. Xie, X. Yu, J. Ge, M. Liu, L. Guo, Preparation of molecularly imprinted polymer microspheres for selective solid-phase extraction of capecitabine in urine samples. Polymers, 14 (2022) 3968. https://doi.org/10.3390/polym14193968.
- Zamruddin, N.M.; Herman, H.; Rijai, L.; Hasanah, A.N. Factors affecting the analytical performance of magnetic molecularly imprinted polymers. Polymers, 14 (2022) 3008. https://doi.org/10.3390/polym14153008.
- Section, which is describing MIP formation and removal of template, could be extended and advanced by adding some more details.
Answer: Thanks for your comment. We have added more details in the revised manuscript. Thanks again for your comment.
- Temperature-controlled release of material is very interesting and promising method, therefore, some discussion on possible multiple reusability of formed MIPs could be added.
Answer: Thanks for your comment. The reusability of formed MIPs was calculated at a RSD of 3.2%, indicating that the prepared MIPs have good reusability.

Round 2
Reviewer 2 Report
Although the Authors have specified in their cover letter that they have addressed all my comments, the changes are not all present in the new version of the paper. I suggest to the Authors to check line by line my previous comments and make the corrections. To identify the sentences that need improvements the Authors need to refer to the original manuscript that was first submitted.
Two of the most important scientific issues that have not been addressed are the following:
- It is still unclear to me whether non-imprinted polymers were really prepared or not. I seem to understand that FMIP without AVAs 2F is a washed MIP and not a NIP. If so, the washing used to remove the template should be clearly specified as well as the analytical technique that was used to assure that the template was fully removed? This is paramount to make the results credible. Why did the Authors decide to use the washed MIP to calculate the IF rather than preparing a NIP? Was the presence of the template affecting the polymerisation significantly and therefore in its absence the resulting NIPs were too different? I would like the Authors to clarify this to me. If a NIP was not prepared, what do the Authors mean with F0 in equation 1? Was this the fluorescence of a fully washed fMIP? Is so this needs to be clarified.
- I have extensive experience in ground materials and I strongly believe that ground polymers cannot result in spherical particles as those showed by the Authors. What type of grinding are the Authors referring to? Again, this needs to be addressed to make the results fully credible.
Author Response
Response to Reviewer 3 Comments
Point1: Although the Authors have specified in their cover letter that they have addressed all my comments, the changes are not all present in the new version of the paper. I suggest to the Authors to check line by line my previous comments and make the corrections. To identify the sentences that need improvements the Authors need to refer to the original manuscript that was first submitted.
Response 1: Thanks for your comment, we have marked these changes in the revised manuscript. Thanks again for your comment.
Point2: - It is still unclear to me whether non-imprinted polymers were really prepared or not. I seem to understand that FMIP without AVAs 2F is a washed MIP and not a NIP. If so, the washing used to remove the template should be clearly specified as well as the analytical technique that was used to assure that the template was fully removed? This is paramount to make the results credible. Why did the Authors decide to use the washed MIP to calculate the IF rather than preparing a NIP? Was the presence of the template affecting the polymerisation significantly and therefore in its absence the resulting NIPs were too different? I would like the Authors to clarify this to me. If a NIP was not prepared, what do the Authors mean with F0 in equation 1? Was this the fluorescence of a fully washed fMIP? Is so this needs to be clarified.
Response 2: Thanks for your comment. The FMIPs without AVAs 2f was not the FNIPs. We used FMIPs to denote the FMIPs load with AVA 2f, and FMIPs without AVAs 2f mean the FMIPs removing targets. We have added the preparation of FNIPs in line 120-121 in the revised manuscript. In addition, IF represented the ratio of Ksv to K*sv. Where Ksv was the slope of the curve between AVAs 2f concentration and FMIPs fluorescence response value, and K*sv was the slope between AVAs 2f concentration and FMIP without AVAs 2f fluorescence response value and the equation was followed at below. F0 was the fluorescence intensity of the material after adding the target (AVAs 2f). Thanks again for your comment.
Point3: - I have extensive experience in ground materials and I strongly believe that ground polymers cannot result in spherical particles as those showed by the Authors. What type of grinding are the Authors referring to? Again, this needs to be addressed to make the results fully credible.
Response 3: Thanks for your comment. Because we had to go through a 200-mesh screen after grinding, and then we got the polymer, and the following references proved this idea. Thanks again for your comment.
- Jw, Y. B. Qin, A. Ny, C. Wh, A. Js, A. Xn, Preparation of multiresponsive hydrophilic molecularly imprinted microspheres for rapid separation of gardenia yellow and geniposide from gardenia fruit, Food Chem. 374 (2021) 131610. https://doi.org/10.1016/j.foodchem.2021.131610.
- Sobiech, J. Giebutowicz, P. Luliński, Application of magnetic core–shell imprinted nanoconjugates for the analysis of hordenine in human plasma-preliminary data on pharmacokinetic study after oral administration, J. Agric. Food Chem. 68 (49) (2020) 14502-14512. https://doi.org/10.1021/acs.jafc.0c05985.
- P. Bagwe, C. Yang, Yang, L. R. Yang, W. Tan, Optimization of dye-doped silica nanoparticles prepared using a reverse microemulsion method, Langmuir. 20 (19) (2004) 8336-8342. https://doi.org/10.1021/la049137j.
- Ren, L. Yang, Y. Li, X. Li, Design and synthesis of a sandwiched silver microsphere/TiO2 nanoparticles/molecular imprinted polymers structure for suppressing background noise interference in high sensitivity surface-enhanced Raman scattering detection, Applied Surface Science, 544 (2021) 148879. https://doi.org/10.1016/j.apsusc.2020.148879.
- Marta, L. Justyna, P. Żaneta, Z. Adriana, The effect of microemulsion composition on the morphology of Pd nanoparticles deposited at the surface of TiO2 and photoactivity of Pd-TiO2, Applied Surface Science, 405 (2017) 220-230. https://doi.org/10.1016/j.apsusc.2017.02.014.
- Irshad Mohiuddin, Aman Grover, Jatinder Singh Aulakh, Sang-Soo Lee, Ashok Kumar Malik, Ki-Hyun Kim, Porous molecularly-imprinted polymer for detecting diclofenac in aqueous pharmaceutical compounds, Chemical Engineering Journal. 382 (2020) 123002. https://doi.org/10.1016/j.cej.2019.123002.
- Betlem, I. Mahmood, R.D. Seixas, I. Sadiki, R.L.D. Raimbault, C.W. Foster, R.D. Crapnell, S. Tedesco, C.E. Banks, J. Gruber, M. Peeters, Evaluating the temperature dependence of heat-transfer based detection: A case study with caffeine and molecularly imprinted polymers as synthetic receptors, Chemical Engineering Journal. 359 (2019) 505-517. https://doi.org/10.1016/j.cej.2018.11.114.
- Liu, X. Dang, H. Ding, H. Chen, Specific recognition and solid phase extraction of three primary aromatic amines based on molecularly imprinted polymer monolith for the migration detection in food contact materials, Microchemical Journal. 182 (2022) 107895. https://doi.org/10.1016/j.microc.2022.107895.
- He, S. Zeng, A.M. Abd El-Aty, A. Hacımüftüoğlu, W. Kalekristos Yohannes, M. Khan, Y. She, Development of water-Compatible molecularly imprinted polymers based on functionalized β-Cyclodextrin for controlled release of atropine, Polymers. 12 (2022) 130. https://doi.org/10.3390/polym12010130.
- Kassem, S. S. Piletsky, H. Yesilkaya, O. Gazioglu, M. Habtom, F. Canfarotta, E. Piletska, A. C. Spivey, E. O. Aboagye, S. A. Piletsky, Assessing the in vivo biocompatibility of molecularly imprinted polymer nanoparticles, Polymers. 14, (2022) 4582. https://doi.org/10.3390/polym14214582.
